Journal of Machine Learning Research 1 (2000) 1-48            Submitted 4/00; Published 10/00

# Multi-scale Regional Attention Deeplab3+: Multiple Myeloma Plasma Cells Segmentation in Microscopic Images

**Editor:**

## Abstract

Multiple myeloma cancer is a type of blood cancer that happens when the growth of abnormal plasma cells becomes out of control in the bone marrow. There are various ways to diagnose multiple myeloma in bone marrow such as complete blood count test (CBC) or counting myeloma plasma cell in aspirate slide images using manual visualization or through image processing technique. In this work, an automatic deep learning method for the detection and segmentation of multiple myeloma plasma cell have been explored. To this end, a two-stage deep learning method is designed. In the first stage, the nucleus detection network is utilized to extract each instance of a cell of interest. The extracted instance is then fed to the multi-scale function to generate a multi-scale representation. The objective of the multi-scale function is to capture the shape variation and reduce the effect of object scale on the cytoplasm segmentation network. The generated scales are then fed into a pyramid of cytoplasm networks to learn the segmentation map in various scales. On top of the cytoplasm segmentation network, we included a scale aggregation function to refine and generate a final prediction. The proposed approach has been evaluated on the SegPC2021 grand challenge and ranked second on the final test phase among all teams.

**Keywords:** Myeloma Plasma Cell, Segmentation, Attention Deeplabv3+, Deep Learning, SegPC2021, Grand Challenge

## 1. Introduction

Cancer happens when the cells start to grow out of control and spread to healthy surrounding tissue. Myeloma, also known as multiple myeloma, is a type of blood cancer that arises from plasma cells in the bone marrow Rajkumar et al. (2014); Guyton and Hall (2006). More specifically, bone marrow is a kind of soft tissue found inside some part of larger bones in the human body. Different types of blood cells such as red blood cells, white blood cells, and platelets are made in the bone marrow Hideshima et al. (2007). Plasma cells developed by the B lymphocytes (type of white blood cells) form part of the body's immune system. To fight infections, antibodies, also known as immunoglobulin, are produced by normal plasma cells. In myeloma cancer plasma cells crow in the bone marrow in a way there is no space for normal red cells, white cells, and platelets. Myeloma begins to develop when the DNA is damaged or changed during the production of new plasma cells. These abnormal plasma cells (myeloma cells) will spread in a different part of bone marrow and produce more abnormal cells. Myeloma cells will produce a large number of paraproteins (type of antibody) which are useless and unable to fight the infections Bird et al. (2011). Unlike other cancers, myeloma will not form a tumor or lump but it will lead to the accumulation of abnormal plasma cells in the bone marrow and paraproteins

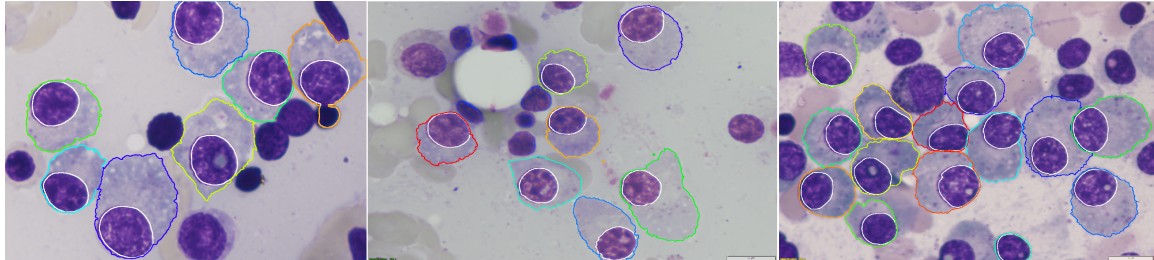

Figure 1: Some samples of myeloma plasma cells microscopy images Gupta et al. (2021), where the myloma cancer cells are detected and highlighted with colored boundary.

in the body. Multiple myeloma is referred to the situation when myeloma cancers affect multiple parts of the body Alexanian and Dimopoulos (1994). Since myeloma cell can be differentiated from normal plasma cells based on histology and morphological features, it is a common method to diagnose multiple myeloma cancer through the aspirate slide images Kyle et al. (2007); Palumbo et al. (2009); Nau and Lewis (2008). In this method, at first, blood samples will be extracted from bone marrow by using the injection of the needle onto the bone. The extracted blood sample will be transferred to a slide and stained using hematoxylin and eosin. Abnormal plasma cells will be detected and marked using manual microscopic visualization (sample is shown in figure 1). Finally, based on the estimation of the normal plasma cells in bone marrow, the presence or absence of myeloma cancer will be concluded Minges Wols (2001).

Although manual inspection of stained slide images is a gold standard of diagnosis of myeloma cancer, it is time-consuming and prone to inter and intraobserver variation. These limitations could be compensated by the use of advanced digital image processing techniques such as object detection and segmentation. Automation of the abnormal plasma cell detection alongside expert pathologist decision could lead to the reduction of diagnosis time and workload of the pathologist. To this end, in this paper, we propose a deep network that utilizes a pyramid of Attention Deeplabv3+ model in a regional-based manner to segment each instance of a myeloma cancer cell. We utilize our approach on the multiple myeloma plasma cell segmentation challenge which provided by Gupta et al. (2018, 2020); Gehlot et al. (2020); Gupta et al. (2021). Our contribution is summarized as follows:

- Second ranking on the SegPC2021 challenge for multiple myeloma cancer cell segmentation.

- Regional base instance segmentation approach.

- guiding cytoplasm segmentation network with additional nucleus mask as a supervisory signal.

## 2. Related Work

Advanced image processing and machine learning methods such as image classification, object detection, and segmentation could have promising applications in various medical domains Azad et al. (2019); Asadi-Aghbolaghi et al. (2020); Feyjie et al. (2020). Detection and segmentation of abnormal cells in microscopic images have been proposed by several researchers in recent works. For instance, Vyshnav et al. used a deep learning-based approach for multiple myeloma cancer detection in stained microscopic images. They compared the performance of Mask R-CNN and U-Net in segmentation and inferred that Mask R-CNN has superior performance than U-Net in myeloma cell segmentation Vyshnav et al. (2020). Authors of Tehsin et al. (2019) used convolutional neural networks to classify the normal and abnormal plasma cells in stained microscopic images. For the classification task, AlexNet Krizhevsky et al. (2012) model is utilized to extract features from microscopic images, and then Support Vector Machine (SVM) applied to the extracted feature set to classify the sample. The main novelty of this work was the pre-processing stage where they used a median filter for each R, G, and B color channel individually and linear contrast stretching for the color enhancement. Vuola et al. used Mask-RCNN and U-Net ensembled for nuclei segmentation in microscopic images. They inferred that Mask-RCNN and U-Net have similar results on the nucleus segmentation task. They reported that U-Net has better performance in nucleus segmentation than Mask R-CNN in the term of similarity index. On the other hand, the Mask R-CNN has better performance in the term of precision assessment. Finally, they concluded that an ensembled model improves the model performance in nucleus segmentation Vuola et al. (2019). Saeedizadeh et. al. Saeedizadeh et al. (2016). used a bottleneck algorithm, modified watershed, and SVM for myeloma cell detection in microscopic images. At first, using the color normalization technique they separated the white blood cell from red blood cells. Then the thresholding technique is used to separate the nucleus from the cytoplasm area. Further, the watershed and bottleneck algorithms are exploited to separate the connected cells. Finally, by using the series of decision rules and the use of an SVM classifier they achieved the sensitivity of 96.52% and precision of 95.28% in recognition of myeloma cells Saeedizadeh et al. (2016). Even though the literature work gained promising results, their applicability in the multiple myeloma cancer cell segmentation is limited due to the challenges such as overlap between cytoplasm's of instances, the fuzzy boundary of the cytoplasm, and overlaying of one nucleus on another cytoplasm in microscopic images. To mitigate these limitations, we propose a regional attention deep model to segment each cell with precise attention.

## 3. Methodology

A general diagram of the proposed structure is depicted in figure 2. The proposed methods consist of two stages: in the first stage, the nucleus segmentation network extracts all the nucleus instances. Then each instance fed into a multiscale cytoplasm segmentation network. This network utilizes the Attention Deeplab3+ model to segment the cytoplasm area. In the next subsections, we will elaborate on each part in more detail.

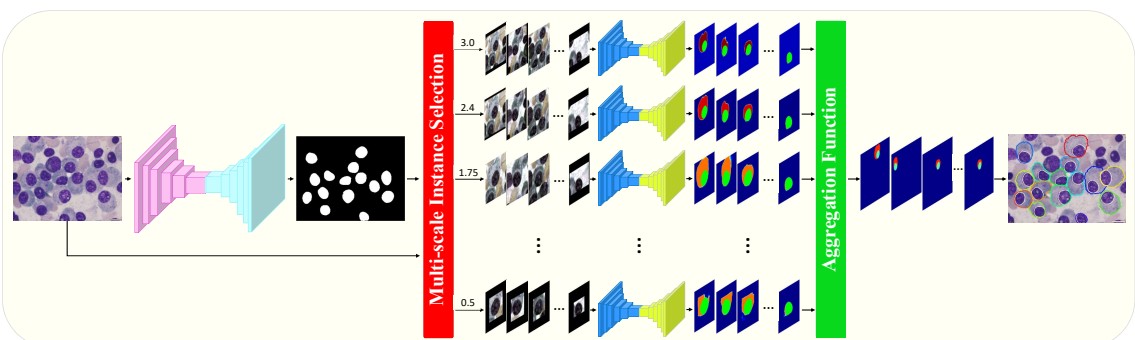

Figure 2: Proposed regional Attention Deeplabv3+ model for multiple myeloma plasma cells segmentation. The proposed method applies a U-net structure to learn the segmentation map for each nucleus instance then it utilizes multi-scale attention deeplabv3+ model to generate the segmentation mask for cytoplasm.

### 3.1 Nucleus Segmentation

In the proposed architecture instead of jointly learning the segmentation of the nucleus and cytoplasm mask, we utilize a two-stage strategy. Our main motivation is to use the detected nucleus instance as a supervisory signal for the cytoplasm instance segmentation to deal with overlapped areas. In other words, the man objective of the first stage is to extract all the possible nucleus instances from the input image. Then each extracted nucleus instance alongside the cropped image patch is fed to the multi-scale instance selection function. The instance selection function is simply an image cropping function with a predefined scale. We use multi-scale to deal with varying cytoplasm scales. In figure 2, a sample of cropped nucleus instances with varying scale sizes (0.5 to 3.0) is demonstrated. To learn the nucleus segmentation map we train a U-net model using a nucleus annotation mask. It is worthwhile to mention that we include the predicted nucleus instance alongside the cropped image as an input for the cytoplasm segmentation network. The goal of this extra input is to guide the network for the object of interest.

### 3.2 Cytoplasm Segmentation

In a regular auto-encoder decoder structure the encoder network consists of several convolutions blocks followed by pooling operations to encode the object of interest in high-level representation space. In this structure, due to the consecutive pooling operation, the spatial dimension of the network may considerably decrease which can result in less discriminated representation power for objects with varying scale. To mitigate this problem, the Deeplabv3+ model utilized an atrous convolution structure. The atrous convolution applies a set of upsampled convolutional kernels to describe the object of interest in higher receptive filed size. To further improve the representation power of the Deeplab model, Azad et al. Azad et al. (2020) proposed a two-level add-on attention mechanism to extract more informative features from the atrous convolutions. Where the first level attention mechanism scales the representation space to highlight the more informative channels then the second

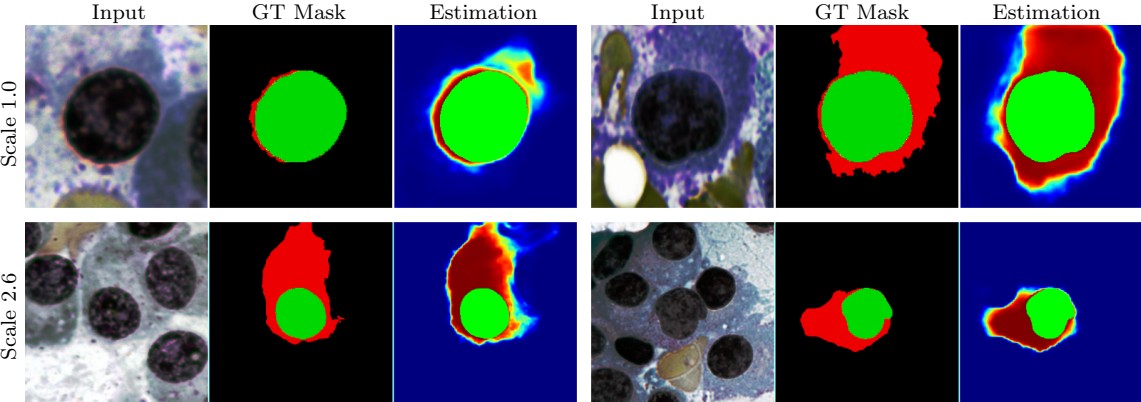

Figure 3: Segmentation results of the proposed method for both nucleus and cytoplasm area. The cropped image alongside the predicted nucleus mask is fed to the cytoplasm network to generate the instance cytoplasm segmentation. Using the detected nucleus mask as a supervisory signal guides the model to separate instances from the highly overlapped background.

attention mechanism utilizes a 3D convolution kernel between each atrous scale to learn a robust non-learn feature set. In this section, we use the Attention Deeplabv3+ model to tackle the cytoplasm segmentation problem. Cytoplasm boundary has a high overlap with the background area and it requires careful attention to discriminate the cytoplasm boundary from the background area. In our implementation, we fed the extracted nucleus area alongside the image batch to the model to learn the instance segmentation mask. We also apply the image histogram equalization method to normalize samples. Sample of the estimated masks for the given nucleus instance is depicted in figure 3.

### 3.3 Aggregation function

Learning objects of interest in multi-scale fashion can produce a robust segmentation mask. In this work, we apply the multi-scale technique on the input level. Consequently, the model generates a multi-scale segmentation mask. The main objective of this multi-scale technique is to tackle the problem of cytoplasm boundary. More specifically, the cytoplasm boundary has a non-rigid shape. If the object boundary (cytoplasm of the detected nucleus) is considerably small, then the models need precise attention around the nucleus boundary with a small scale. On the other hand, if the boundary is big then the model needs big attention to separate it from other instances. We solve this limitation by defining several scales. Since the ultimate objective of the model is to produce a single segmentation mask for each instance, we propose to use an aggregation function to combine and select a single segmentation mask. The aggregation function can use the output of all scales to generate a single prediction (like non-maxima suppression), however, in our experiment we observe that selecting a scale for each instance can produce better performance than non-maxima suppression. To perform this operation, we simply start from the lowest scale and calculate the relation between the detected cytoplasm area and the nucleus area. If the ratio is higher

than a threshold value then the next scale is evaluated. The process goes through the next scales until finding the appropriate condition. In other words, how local the network should focus to separate the object from the background and other instances.

### 3.4 Training Procedure

Our training procedure consists of two stages. In the first stage, we train the U-net model to segment the nucleus from the input images. The training process takes into account the training and validation set and learns the nucleus mask. We train the model for 100 epochs using the Adam optimization with a learning rate of $1e - 4$. In the second phase, we trained each Attention Deeplabv3+ model using the patches extracted from the input image alongside the nucleus mask (resulted from the nucleus segmentation network). For each scale, we train the model for 100 epochs using the Adam optimization with a learning rate of $1e - 5$. All training is done using cross-entropy loss on a single GTX 1080 GPU.

### 3.5 Inference Procedure

The inference stage uses the trained models to generate the segmentation mask for both nucleus and cytoplasm instances. In our inference, we use a fixed number of scales (4 scales) for the test phase.

## 4. Results

The proposed method is evaluated on multiple myeloma cell segmentation grand challenges which are provided by the SegPC 2021. The challenge data set consists of a training set with 290 samples, validation and test sets with 200 and 277 samples respectively. All the samples are annotated by the pathologist and instance base segmentation masks are provided for the object of interest (myeloma plasma cells). We trained our model using the training and validation set. During the competition time, we generated the segmentation mask for each instance. The challenge leaderboard compared each team using the MIOU metric, where our method ranked second among all teams. Table 1 shows the comparison results for the top five winning teams.

Table 1: Performance comparison on the final test phase for SegPC2021 grand challenge seg

| Teams | Ranking | Score (mIoU) |
|---|---|---|
| XLAB Insights | 1 | 0.9389 |
| **bmdeep (Proposed Method)** | 2 | **0.9385** |
| DSC-IITISM | 3 | 0.9382 |
| 507 | 4 | 0.9366 |
| AIVIS | 5 | 0.9276 |

As shown in table 1, the proposed method outperformed most of the competitors and achieved the second-best place with a small gap (0.0004) from the first team. It is worthwhile to mention that the Xlab approach combined several instances of state-of-the-art

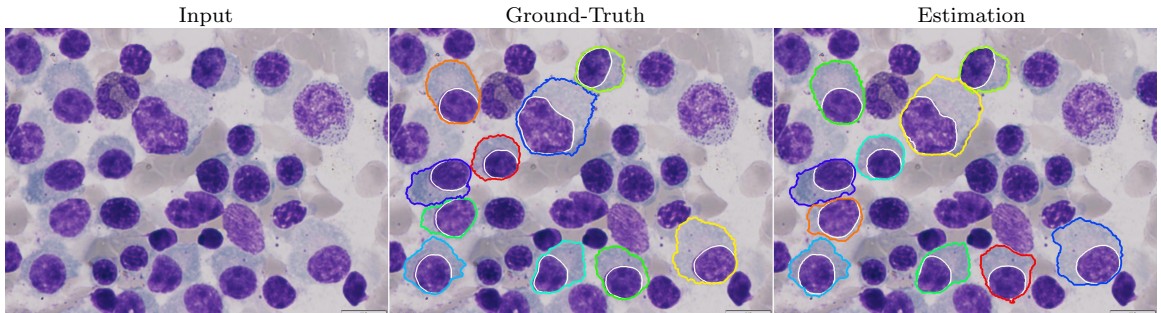

Figure 4: A sample of prediction results on SegPC2021 grand challenge.

segmentation architectures such as SCNet Vu et al. (2021) and ResNeSt Zhang et al. (2020) with minor modifications to suit them for the cell segmentation task. In addition, heavy image augmentation has been reported which leads to enhance the performance of the model. While this ensemble approach slightly increased the performance, its computational and memory complexity makes it less effective for real-world application. On the other hand, our proposed method comparatively requires less memory and fewer inferences time on a single GPU. We would like to point out that both third and fourth teams utilized Mask-RCNN for cell segmentation. This approach is well known for both instance segmentation and object detection. In the other words, it is capable of separating the object in the image using its corresponding mask and bounding box. This architecture relies on region proposals (generated by region proposal network) and a feature extractor. The region proposal network will be followed by an ROI alignment operation which produces desirable output for the region classifier. Finally, this model uses a fully convolution network for instance segmentation and a region classifier for class prediction. The training process of this network involves minimizing multiple loss functions for several task learning. Although Mask-RCNN suits well for the instance segmentation task, it fails to produce a precise segmentation mask for low resolution and highly overlapped objects. While our proposed method uses the predicted nucleus mask as a supervisory signal to overcome the highly overlapped objects. Furthermore, Due to the invariant sailing of the fully convolutional neural network structure of Mask-RCNN, this network is not capable to differentiate the spatial information between different receptive fields of different sizes. Thus, Semantic information resulted from small-scale receptive fields, will have less capability to capture the object boundary with varying shapes. To overcome these issues, our method utilizes a multi-scale learning strategy to to decrease the effect of scales variation and learn the shape variations through multi-scale representation. Figure 4 demonstrates some prediction results where the proposed method estimated both nucleus and cytoplasm masks with high performance.

As explained earlier the proposed method uses a multi-scale strategy to generate a precise segmentation mask for the cytoplasm instances. In this section, we will elaborate on the scale selection strategy and its effect on the final performance. To this end, we have extracted the statistical scale information from the training set. The histogram information is depicted in figure 5.

According to figure 5, we can observe that the ratio of cytoplasm area to nucleus area is distributed in almost four different peaks (shown with green circles). Thus, we select four

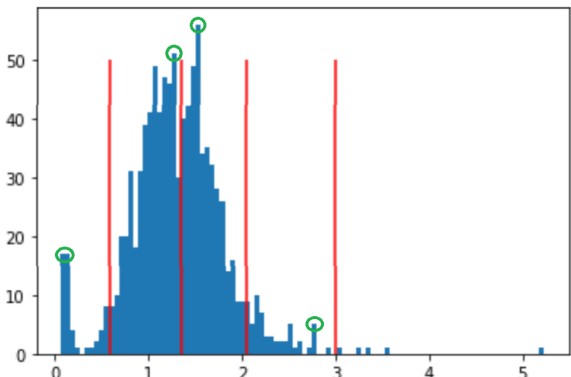

Figure 5: Histogram of the area of cytoplasm to the nucleus on the validation set. Histogram peaks are highlighted with green circles to show the importance scale values. slightly bigger values selected for each peak (shown with red line) to produce a better receptive field size.

scales to generate a precise segmentation mask. It is worthwhile to mention that we select the scale value a bit higher than the histogram peaks (shown with red line on figure 5) to generate an image patch to cover the appropriate receptive field size. In our experiment for the final test phase, we selected 4 different scales as depicted in figure 2. Overall, using each scale separately can produce a 92.5 mIoU. hence we used the aggregation technique to boost the performance.

## 5. Conclusion

In this paper, we proposed a multi-scale regional Attention Deeplabv3+ model for myeloma plasma cell segmentation. The proposed method utilized a U-net model for nucleus instance segmentation. The segmented nucleus instance is extracted from the input image and alongside the predicted nucleus mask fed into a multi-scale cytoplasm detector network. The cytoplasm detector took into account the strength of the Attention Deeplabv3+ model to segment each cytoplasm instance. We further proposed an aggregation function to select the more related scale to fulfill the prediction score. Evaluation results on the final challenge phase demonstrated outstanding results.

**Acknowledgements**. All the implementation code is available:
https://github.com/bmdeep/SegPC2021

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
