# OpenReview forum: "Multi-scale Regional Attention Deeplab3+: Multiple Myeloma Plasma Cells Segmentation in Microscopic Images"
_MICCAI.org/2021/Workshop/COMPAY — COMPAY 2021_

### Official Review · Reviewer_Q5jN · 2021-08-06
**??**

**Rating:** 7
**Confidence:** 3

**Review:**

The paper presents an approach to detect and segment plasma cells for diagnosis of multiple myeloma cancer. The abnormalities can display a range of sizes relative to the nucleus. The authors propose a multiscale approach to improve segmentation. The scales used are derived from the distribution of sizes in the training data. The method is applied to the SegPC 2021 challenge and ranked second.

The results are promising and the approach, although based on a previously introduced network, contains interesting ideas for this application. What would improve the work is a better study of the added value of the multiscale method. Looking at the distribution of ratios, the range of scales seems limited. The vast majority of cases lie within scales 1 tot 2. It would be interesting to see what the outcomes are without the multiscale approach. Also, the choice of scales is somewhat ad hoc. This is  not unusual in a first conference paper, but some evaluation of the sensitivity of the method to this choice should be performed. This could be future work, although it is not a huge effort to include it now.

The paper is not properly balanced. It contains few details on the method and a lengthy discussion comparing the proposed method to those of other contestants in the challenge. A short description of the data should be included. The explanation of the scale selection should be presented prior to the results.

In the introduction the authors state that their method deals with various limitations of state-of-the-art methods, such as overlapping structures and fuzzy boundaries. The paper does not show that the proposed method does indeed perform better for such instances.

In 3.1 you refer to figure 2. This should probably be figure 3, although that figure shows examples at different scales than indicated in the text.
The list of references is rather extensive for a conference paper.

---

### Official Review · Reviewer_sK2V · 2021-08-20
**Multi-scale extension of previous published methods with good results on a challenge dataset**

**Rating:** 7
**Confidence:** 5

**Review:**

The paper presents a nuclei and cell segmentation method for myeloma plasma cells. The proposed method builds on previous work (attention Deeplab3+ model) by adding a multi-scale component that addresses the issue of varying size of the plasma cells.

The manuscript is generally clearly written but the methodology should be better explained (in particular, the aggregation method is not very clear while it is an essential component). It would be also interesting to see a comparison to other aggregation methods. The proposed methodology is well motivated and it seems suitable for this particular task.

It is unclear if the nuclei segmentation that was used as an additional supervision for the trained model was part of the challenge dataset or was additionally produced by the authors. Please clarify.

Minor remarks:
- There are occasional typos: "crow", "non-learn feature set".
- The references are not in correct format, all seem to be in-line citations which is a bit distracting when reading.

---

### Decision · Program_Chairs · 2021-08-25

Accept